# Hemin-Induced Endothelial Dysfunction and Endothelial to Mesenchymal Transition in the Pathogenesis of Pulmonary Hypertension Due to Chronic Hemolysis

**DOI:** 10.3390/ijms23094763

**Published:** 2022-04-26

**Authors:** Janae Gonzales, Kelsey Holbert, Kamryn Czysz, Joseph George, Caroline Fernandes, Dustin R. Fraidenburg

**Affiliations:** 1Department of Medicine, University of Illinois at Chicago, Chicago, IL 60612, USA; jgonz93@uic.edu (J.G.); jgeorg36@uic.edu (J.G.); csferna2@uic.edu (C.F.); 2Department of Medicine, University of Wisconsin, Madison, WI 53792, USA; kholbert@uwhealth.org; 3Department of Chemistry, University of Denver, Denver, CO 80208, USA; kamryn.czysz@du.edu

**Keywords:** endothelial dysfunction, endothelial to mesenchymal transition, pulmonary hypertension, hemolysis

## Abstract

Pulmonary hypertension in sickle cell disease is an independent predictor of mortality, yet the pathogenesis of pulmonary vascular disease in chronic hemolytic disorders remains incompletely understood and treatment options are limited primarily to supportive care. The release of extracellular hemoglobin has been implicated in the development of pulmonary hypertension, and in this study we explored the direct effects of hemin, the oxidized moiety of heme, on the pulmonary artery endothelium. We found that low dose hemin exposure leads to significantly increased endothelial cell proliferation, migration, and cytokine release as markers of endothelial dysfunction. Protein expression changes in our pulmonary artery endothelial cells showed upregulation of mesenchymal markers after hemin treatment in conjunction with a decrease in endothelial markers. Endothelial to mesenchymal transition (EndoMT) resulting from hemin exposure was further confirmed by showing upregulation of the transcription factors SNAI1 and SLUG, known to regulate EndoMT. Lastly, given the endothelial dysfunction and phenotypic transition observed, the endothelial cytoskeleton was considered a potential novel target. Inhibiting myosin light chain kinase, to prevent phosphorylation of myosin light chain and cytoskeletal contraction, attenuated hemin-induced endothelial hyper-proliferation, migration, and cytokine release. The findings in this study implicate hemin as a key inducer of endothelial dysfunction through EndoMT, which may play an important role in pulmonary vascular remodeling during the development of pulmonary hypertension in chronic hemolytic states.

## 1. Introduction

Pulmonary hypertension (PH) is increasingly recognized as a critical complication of sickle cell disease (SCD). Nearly 10% of individuals with SCD develop PH, which is an independent predictor of mortality with an associated median survival limited to 6.8 years after time of diagnosis [1]. Approximately half of these patients have precapillary PH, sharing key pathobiological features with Group 1 pulmonary arterial hypertension (PAH), and the remaining individuals have evidence of chronic thromboembolic PH (CTEPH) or postcapillary PH related to left heart dysfunction [2].

The etiology of PH in this population is poorly understood, but due to the increased incidence of PH in other chronic hemolytic disease states, such as thalassemia, hereditary spherocytosis, and paroxysmal nocturnal hemoglobinuria, it is thought to be a consequence of intravascular hemolysis [2]. During hemolysis, red blood cells undergo rapid erythrocyte lysis, overwhelming mechanisms of normal red cell turnover at several protein binding sites, and leading to the release of free heme [3]. The detrimental effects of hemolysis cause both acute and chronic adverse effects across several organ systems including the heart, lungs, liver, and kidneys, and many studies have implicated the toxicity of extracellular heme and its permeation of different cell types as the inciting trigger [4,5,6,7,8,9,10]. In SCD cohorts, prior studies have identified increased concentrations of free hemoglobin and hemin, the oxidized moiety of heme, in the steady state plasma of individuals with SCD when compared to individuals without [11,12]. Additionally, these elevations directly correlate with clinical markers of hemolysis, incidence of PH, and risk of mortality [13]. The current mechanistic understanding of how free heme and byproducts of hemolysis lead to the development of PH is via extracellular heme’s propensity to scavenge nitric oxide, thereby reducing its bioavailability and favoring a vasoconstrictive phenotype that results in early precapillary PH [14]. Treatment of SCD-related PH is limited by the incomplete understanding of the pathways that connect hemolysis with the development of PH. Current guidelines for SCD-related PH focus on SCD therapies that are beneficial in all high risk sickle cell disease patients, but only indirect evidence links these therapies to PH. Hydroxyurea increases fetal hemoglobin and is associated with improved overall survival in SCD, in which a majority of deaths arise from pulmonary complications [15]. Chronic transfusion therapy is also advocated by current guidelines in order to reduce the percentage of sickle hemoglobin in patients that are unable to tolerate hydroxyurea [15]. Both of these therapies target decreased hemolysis as the major mechanism of therapeutic benefit, by either increasing fetal hemoglobin or diluting sickle hemoglobin paired with increased hemoglobin A. The impact of these treatments on the pulmonary vasculature, as well as mechanisms of hemolysis leading to the development and progression of pulmonary hypertension, remains unclear.

The pulmonary endothelium serves several functions and is considered quiescent in health, but once activated is shown to contribute to many disease states [16]. Endothelial activation and dysfunction have been identified as early steps that lead to the vascular remodeling seen in PAH [17]. Endothelial dysfunction characterized by disorganized hyperproliferation of pulmonary artery endothelial and smooth muscle cells results in obstruction of vessel lumens, increasing the pulmonary vascular resistance that ultimately leads to right heart failure in individuals with PH [16,18,19]. Endothelial dysfunction also alters the balance of endothelium-derived vasoactive mediators and is an important therapeutic target in our current treatment for PH. Several processes have been recognized as inducers of endothelial dysfunction, including the phenotypic changes that characterize endothelial to mesenchymal transition (EndoMT) [1,20,21].

EndoMT is a mechanism of cellular transdifferentiation involved in several malignant, fibrotic, inflammatory, infectious, and vascular human diseases [22]. In this process, endothelial cells progressively decrease their cell specific proteins and gain mesenchymal characteristics. Ultimately, the cells lose their endothelial barrier protective function and gain the morphology of mesenchymal cells along with their cellular motility and contractile properties [22]. The molecular and regulatory mechanisms of EndoMT have not been fully elucidated, however several cellular signaling pathways and inducers have been identified in relation to embryogenesis and human pathologies [23,24]. This mechanism has been associated with diseases by the identification of an intermediate EndoMT phenotype represented by the co-expression of both endothelial and mesenchymal markers. Specifically, EndoMT has been implicated in PAH with notions that the intimal proliferation in small and medium pulmonary arteries may be a result of an EndoMT process [25,26]. Studies have found evidence of EndoMT in pulmonary arteries of patients with primary PAH and precapillary PH associated with idiopathic pulmonary fibrosis and systemic sclerosis [27,28].

In the current study, we explore the effects of hemin, the oxidized moiety of heme, on pulmonary artery endothelial cells. We hypothesize that hemin exposure will lead to the characteristic endothelial dysfunction that precedes vascular remodeling, and we hypothesize that EndoMT is the mechanism mediating the endothelial dysfunction induced by hemin exposure.

## 2. Results

### 2.1. Hemin Downregulates Endothelial Marker Expression and Upregulates Mesenchymal Marker Expression in Pulmonary Artery Endothelial Cells

Endothelial to mesenchymal transition (EndoMT) has been implicated as a mechanism for the endothelial dysfunction that leads to vascular remodeling seen in PAH [29]. This phenomenon occurs when endothelial cells lose their endothelial characteristics and begin to develop a mesenchymal phenotype. Steady state free heme in SCD approximates hemin ranges from negligible to 20 µM with an average of 4 µM, assuming complete conversion to hemin [11]. Early data had suggested that the peak effect of hemin on our endothelial cells occurred at a dose of 5 µM (Appendix A). To understand if hemin-treated cells undergo EndoMT, western blot analysis was used to probe for both endothelial and mesenchymal cell markers. As shown in Figure 1A, HPAECs treated with 5 µM hemin exhibited a relative decrease in endothelial cell protein expression over time of von Willebrand factor and the adhesion molecule vascular endothelial (VE) cadherin, when compared to control. Coupled with decreased expression of endothelial markers, hemin-treated cells acquire expression of mesenchymal proteins vimentin and fibronectin which shows a relative increase over time as compared to control conditions. VE-cadherin had large fluctuations in expression in these transitional HPAECs with a non-significant trend in reduced expression at 24 h after 5 µM hemin treatment (Figure 1B, *p* = 0.11) while the mesenchymal marker α-smooth muscle actin (αSMA) is significantly increased (Figure 1C, *p* < 0.05). The presence of both endothelial cell and mesenchymal cell markers is consistent with EndoMT and a transitional cell phenotype. To further confirm if hemin promotes EndoMT in HPAECs, immunofluorescence was utilized to examine expression of α-smooth muscle actin in these cells. In Figure 1D, HPAECs treated with hemin demonstrated increased expression of αSMA when compared to control. We also chose to evaluate the use of a myosin light chain kinase (MLCK) inhibitor (ML-7) on hemin-treated HPAEC, as cytoskeletal reorganization is an established feature of EndoMT [30,31]. Pre-treatment with ML-7 (20 μM) prevented the increased αSMA expression that was seen in hemin-treated endothelial cells. Taken together, the decreased endothelial marker expression and increased mesenchymal marker expression is consistent with phenotypic changes of EndoMT.

### 2.2. Hemin Upregulates EndoMT Transcription Factors 

To validate that this upregulation of mesenchymal phenotype with downregulation of endothelial-specific cell markers is mediated by an EndoMT mechanism, hemin-treated cells were probed for expression of established EndoMT transcription factors, zinc finger proteins SNAI1 and SLUG (SNAI2). In Figure 2, SNAI1 and SLUG expression is upregulated in a time-dependent manner in hemin-treated cells when compared to control. Protein expression is significantly increased in hemin-treated HPAEC at 8 h for both SNAI1 (Figure 2B) and SLUG (Figure 2C). This data shows that, in addition to characteristic transitional phenotype changes in hemin-treated HPAEC, EndoMT transcription factors are also upregulated.

### 2.3. Hemin Induces Endothelial Cell Viability and Proliferation Which Is Prevented by a Myosin Light Chain Kinase Inhibitor

Since hemin treatment in HPAECs led to evidence of EndoMT, we sought to evaluate for the endothelial cell dysfunction that is a consequence of EndoMT and leads to pulmonary vascular remodeling in PH [32]. Endothelial cell dysfunction in precapillary PH is characterized by an uncontrolled proliferation that leads to vascular remodeling and formation of plexiform lesions [32]. We confirmed these across several experiments by treating HPAECs with 5 µM hemin for 24 h, as this specific time point demonstrated the greatest characteristics of EndoMT. HPAECs were then subject to measurements of cell viability by water soluble tetrazolium salt 1 (WST-1) assay and proliferation by protein expression of proliferating cell nuclear antigen (PCNA) as a component of the cell replication machinery. In Figure 3A, HPAECs treated with hemin demonstrated more than a 50% increase in viability at 24 h when compared to control (*p* < 0.001). Hemin-treated HPAECs also had increased PCNA expression nearly three-fold after 24 h of treatment when compared to control (Figure 3B,C; *p* < 0.05). Based on this data, hemin treatment is associated with increased endothelial viability and hyperproliferation. To further elucidate the role of HPAEC cytoskeletal machinery as an important feature and potential therapeutic target of EndoMT leading to endothelial dysfunction, an MLCK inhibitor was used to prevent MLC phosphorylation and actin-myosin cross-linking [33,34,35]. In Figure 3, ML-7 pre-treatment decreases viability and proliferation when compared to hemin-treated cells by 33% in WST assay (Figure 3A; *p* < 0.001) and decreased PCNA expression by 87% (Figure 3B,C; *p* < 0.05).

### 2.4. Hemin Induces Cytokine Release Which Is Prevented by MLCK Inhibition

We evaluated further known characteristics of endothelial dysfunction, such as increased cytokine release leading to a pro-inflammatory state. To determine whether hemin induces inflammation, cytokine production of interleukins IL-6 and IL-8 was evaluated by ELISA technique. IL-6 and IL-8 are known to play important roles in PAH and are also upregulated in in vitro models of induced EndoMT PAECs. [21,35]. As shown in Figure 4A, HPAECs treated in hemin for 24 h had a nearly 50% increase in IL-6 production when compared to control (*p* < 0.001). Similarly, IL-8 production was increased in hemin-treated cells by 35% (Figure 4B; *p* < 0.001). We show here that hemin treatment is associated with endothelial dysfunction and pro-inflammatory cytokine release. Additionally, ML-7 pre-treatment decreased IL-6 secretion by more than 75% (Figure 4A; *p* < 0.001) and IL-8 secretion by 48% (Figure 4B; *p* < 0.001) at 24 h when compared to hemin-treated cells alone. 

### 2.5. Hemin Induces Endothelial Cell Migration and Myosin Light Chain Kinase Inhibitor Negates the Hemin-Induced Hyper Migration

The process of vascular remodeling in the development and progression of PAH is also mediated by increased and disorganized cellular migration [16]. Cellular migration is measured after disrupting the cell monolayer and measuring the recovery and regeneration of the monolayer. Cells were treated with hemin 5 µM and then subsequently subjected to ECIS-based wounding. As shown in Figure 5A, hemin-treated cells demonstrated increased transendothelial resistance (TER) after wounding, indicating a faster recovery when compared to control based on multiple comparison of the continuous data (*p* < 0.05). The area under the curve measurements for these conditions show a nonsignificant trend toward increased migration (Figure 5B; *p* = 0.144). Pre-treatment with ML-7 significantly decreased the TER measured by ECIS after analysis of the continuous data (Figure 5A; *p* < 0.001) and area under the curve (Figure 5B; *p* < 0.05). 

In addition to ECIS analysis, cellular migration was also evaluated by traditional scratch assay to visualize wound closure. In Figure 6A,B, hemin-treated endothelial cells show a trend toward increased migration over the created wound in comparison to control at 4 h (26% versus 19% in hemin-treated and control, respectively, *p* < 0.01) and 8 h (50% vs. 34%, *p* < 0.05) time points. ML-7 pre-treatment in hemin-treated cells was also associated with prolonged wound gap closure time in scratch assay at 4 h (26% versus 18%, *p* < 0.05) and 8 h (50% vs. 28%; *p* < 0.01) (Figure 6). Together, these data suggest that hemin treatment is associated with a relative increase in endothelial cell migration and that inhibition of MLCK can prevent the increased migration that characterizes endothelial dysfunction in hemin-treated HPAEC.

## 3. Discussion

Hemolysis-associated endothelial dysfunction has been a suggested mechanism of pulmonary hypertension (PH) in sickle cell patients due to a similar disease burden of PH among other chronic hemolytic disease states, such as thalassemia and paroxysmal nocturnal hemoglobinuria [2]. Prior studies in sickle cell patients and murine models have demonstrated a relationship between the breakdown products of hemolysis and the development of pulmonary hypertension largely associated with dysregulation of nitric oxide [14,36]; however, the mechanisms remain incompletely understood. In this study, we found that low dose hemin exposure in HPAEC induces endothelial dysfunction characterized by increased proliferation and inflammation. These changes are consistent with early events, leading to vascular remodeling in the development and progression of pulmonary arterial hypertension. Endothelial to mesenchymal transition is identified as a potential mechanism contributing to this endothelial dysfunction seen in hemin-treated cells.

This study augments current understanding of hemolysis-associated pulmonary hypertension across several dimensions. Steady state plasma studies have identified increased concentrations of free hemoglobin and hemin in individuals with sickle cell disease [11,12]. Additionally, these elevations directly correlate with clinical markers of hemolysis, incidence of pulmonary hypertension, and risk of mortality [13]. The relationship between precapillary PH and hemolysis outside of chronic hemolytic disorders has been newly explored in the literature. A recent study demonstrated that the Sugen/hypoxia rat model as well as individuals with PAH have higher steady state plasma free hemoglobin which correlated with disease severity [37]. Furthermore, our current mechanistic understanding of how free heme and the byproducts of hemolysis lead to the development of PH is via extracellular heme’s propensity to scavenge nitric oxide, thereby reducing its bioavailability and favoring a vasoconstrictive phenotype that results in early precapillary PH [14]. In this study we also show direct effects of the breakdown products of chronic hemolysis (hemin) on HPAEC through the novel mechanism of EndoMT, leading to the endothelial dysfunction.

Experimental administration of hemin in sickle cell mice has also been studied, leading to the implication of hemin in the pathogenesis of acute chest syndrome (ACS). Ghosh and colleagues found that administration of hemin leads to lethal acute lung injury in sickle cell mice, and that this effect is reversed with heme-binding protein, hemopexin pre-treatment [38]. Additionally, in vitro studies on hemin-treated lung microvascular endothelial cells demonstrated endothelial barrier dysfunction via necroptosis in a concentration-dependent manner [39]. The pathologic discordance of findings between the ACS studies and the current study can be explained by the concentration of hemin used in these experiments. The ACS in vitro studies used 40 µM hemin and in vivo studies used concentrations of 0.43 mM, representing the large acute hemolytic reaction expected from ACS [38,39]. From a translational perspective, free hemoglobin in SCD patients with lethal ACS approximate 2.23 mM hemin concentration [38,40]. Steady state free heme in SCD approximates hemin ranges from negligible to 20 µM with an average of 4 µM, assuming complete conversion to hemin [11]. This study’s concentration of 5 µM hemin is therefore a plausible chronic concentration in sickle cell patients. From prior studies of hemin in ACS and our current study, we can infer that the concentration of hemin is a significant factor, and further studies exploring different doses for different durations may improve understanding of the relationship of hemolysis timing and severity within different pulmonary complications of sickle cell disease. Within this study alone we found significant variability with small µM fluctuations in the dose of hemin, indicating a narrow window of effect (Appendix A). Evidence suggests that the cellular response to hemin is elicited through toll-like receptor 4 (TLR4) and downstream effects of TLR4 activation vary based on the particular ligand or molecule [39,41,42].

This is the first study to implicate EndoMT in the mechanism of pulmonary hypertension associated with chronic hemolysis and provides a potential mechanism for the vascular remodeling that is found in patients with precapillary PH and sickle cell disease. Expression of αSMA in pulmonary artery endothelial cells undergoing EndoMT is a regular finding in the remodeled arteries of patients and murine models with PAH [29]. While our current work does not specifically implicate interaction with particular cell surface receptors, activation of TLR4 has been previously associated with promotion of EndoMT [43]. This study not only shows that hemin induces the representative findings of downregulated endothelial characteristics, acquisition of mesenchymal markers, and upregulation of EndoMT transcription factors, but also mirrors the endothelial properties of an induced EndoMT HPAEC described in PAH previously. Good et al. demonstrated the presence of EndoMT in lungs and murine models of PAH, but also evaluated the in vitro function of induced EndoMT HPAECs [27]. They found that EndoMT demonstrated increased cellular migration and increased secretion of proinflammatory cytokines, such as IL-6 and IL-8, reflective of the hemin-induced function of HPAECs in this study. IL-6 and IL-8 are known to play important roles in PAH and are associated with increased mortality in PAH patients [44]. Interestingly, circulating IL-8 has been shown to be upregulated in sickle cell disease patients and is thought to have an important role in the pathogenesis of this disease [45,46]. Similarly, IL-6 has been implicated as a marker of endothelial activation in vascular inflammation of transgenic sickle cell mice and vaso-occlusive crises of sickle cell disease patients [47,48]. A recent study employing sickle cell mice is consistent with some aspects of this study’s current findings. Gbotosho and colleagues administered plasma heme in Townes sickle cell mice that resulted in increased IL-6 levels and increased gene expression of cardiac hypertrophy, consistent with the notion that increased hemolysis in sickle cell disease leads to an inflammatory-mediated vasculopathy [3]. Our current study is limited by in vitro analysis; validation with in vivo models demonstrating EndoMT and the development of PH after administration of hemin in future studies will help to strengthen these findings.

Given the implication of EndoMT as a mechanism for hemin-induced endothelial dysfunction, we felt that targeting the cellular cytoskeleton would be a novel and innovative approach to the transforming phenotypic characteristics of these cells. Cytoskeletal rearrangement is a key feature of EndoMT which is thought to contribute to changes in cell shape, invasiveness, and migratory properties [30]. Non-muscle MLCK is an important component of the cellular cytoskeleton which is shown to contribute to normal cell shape and barrier function of HPAECs. Several studies have demonstrated that phosphorylation of myosin light chain (MLC) leads to increased cytoskeletal tension, cellular rounding, and paracellular gap formation resulting in endothelial hyperpermeability [49,50,51]. Although this study is looking at different classifications of endothelial dysfunction, such as disorganized hyperproliferation and increased cellular migration, MLCK through ERK signaling has also been implicated in these mechanisms seen with breast cancer cells, and MLCK’s specific inhibitor ML-7 attenuates this effect [52]. In the current study, ML-7 similarly abrogates the hemin-induced endothelial dysfunction by reducing increased proliferation, cytokine release, and cellular migration. The current study also supports the notion that endothelial cells are undergoing a cellular transdifferentiation process and developing a mesenchymal phenotype. In mesenchymal cells, MLCK plays an important role in smooth muscle cell and fibroblast migration [53,54]. Studies have also shown that decreases in MLCK expression is associated with fibroblast rounding, decreased cellular proliferation, and reduced motility [55]. We also speculate that some of the protective effects seen from ML-7 on hemin-treated cells may be due to MLCK’s aforementioned effects on the newly transitioned mesenchymal properties. A study investigated MLCK inhibition with ML-7 on rat pulmonary artery smooth muscle cells and found that ML-7 treatment reduced proliferation in vitro and reduced both hemodynamic and vascular remodeling markers in a preclinical PAH murine model [56]. Additional work should examine properties of the endothelial cytoskeleton during EndoMT to further implicate this mechanism in pulmonary vascular remodeling and pulmonary hypertension. Interrupting the early stages of EndoMT and its effects on the cellular cytoskeleton may be a novel approach to treating hemolysis-associated pulmonary hypertension.

## 4. Materials and Methods

### 4.1. Reagents

Hemin (catalog# 51280) was obtained from Sigma-Aldrich (St. Louis, MO, USA) and prepared as previously described [39]. Hemin powder was dissolved in 20 mM NaOH and HCl was slowly added to decrease the pH to 7.4. This solution was diluted in endothelial growth media obtained from Lonza (Basel, Switzerland) to obtain the desired concentration for experiments. MLCK-inhibitor, ML-7 (catalog# I2764), was also obtained from Sigma-Aldrich and powder was dissolved in DMSO to create a stock solution. The solution was diluted in endothelial growth media to obtain the desired concentration for experiments. For all individual experiments that used ML-7, pre-treatment occurred for 1 h prior to the addition of described treatment conditions. Normal human pulmonary artery endothelial cells (HPAECs; catalog# CC-2530) and Endothelial Growth Media-2 (EGM-2) were obtained from Lonza. Cell proliferation assay kit (catalog# 2210) was obtained from Millipore (Burlington, MA, USA), Human Interleukin-6 (IL-6) ELISA kit (catalog# ab46042) was obtained from Abcam (Cambridge, UK), and Human Interleukin-8 (IL-8) ELISA kit (catalog# EZHIL8) was obtained from Millipore. Anti-proliferating cell nuclear antigen (Anti-PCNA) antibody (catalog# sc-56) was obtained from Santa Cruz Biotechnology, Inc. (Dallas, TX, USA). Anti-α-smooth muscle actin (catalog# A2547), anti-fibronectin (catalog# F7387), and anti-vimentin (catalog# V6630) antibodies were obtained from Sigma-Aldrich. Anti-von Willebrand factor (catalog# AB7356), Anti-SNAI1 (Snail homolog 1; catalog# MABE167), and Anti-SLUG (SNAI2; Snail homolog 2; catalog# ABE993) antibodies were obtained from Millipore. Anti-vascular endothelial-cadherin (VE cadherin; catalog# D87F2) antibody was obtained from Cell Signaling Technology (Danvers, MA, USA).

### 4.2. Cell Culture

Normal HPAECs were cultured at passages 5–8 using EGM-2 containing 10% fetal bovine serum (FBS) and supplemental growth factors according to the manufacturer’s instructions at 37 °C in a humidified 5% CO_2_ incubator. For all experiments, once cells reached confluence, growth media was completely removed, and cells were serum starved overnight with EGM-2 containing 0.1% FBS. Subsequently, serum starved media was removed and replaced with EGM-2 containing 2% FBS with added treatment or control conditions as described per individual experiment.

### 4.3. Cell Proliferation and Viability Assay

HPAECs were treated with lower concentrations of hemin ranging from 5 to 20 µM for 24 h and subsequently measured cellular proliferation and viability via WST-1 assay (Millipore, Burlington, MA, USA) and proliferation via western blot measurements of proliferating cell nuclear antigen (PCNA) protein expression as a component of the cellular replication machinery. As shown in Appendix A, cellular viability is increased with hemin treatment compared to control, with the most significant effect noted at 5 µM. In Appendix A, PCNA expression is increased compared to control at 5 µM and 10 µM concentrations, with the greatest response at 5 µM which was the concentration chosen for further experiments. 

25 × 10^4^ cells were plated on a 96-well microplate and cultured for 24 h prior to undergoing treatment with 5 μM hemin, PBS, or 5 μM hemin and 20 μM ML-7 for 24 h. Cell viability and proliferation were measured using commercially available WST-based cell proliferation assay kit from Millipore, according to the manufacturer’s instructions. Microplate reader was used to measure absorption at wavelength 450 nm and reference wavelength of 650 nm. Data presented as mean ± standard error (SE), with *n* defined as each independently treated well for a given condition.

### 4.4. Enzyme Linked Immunosorbent Assay (ELISA)

HPAECs were grown to confluence on 6-well plates and treated with 5 μM hemin, PBS, or 5 μM hemin and 20 μM ML-7. After 24 h, the cell culture media was removed and centrifuged, and the supernatant was used for ELISA experiments. IL-6 and IL-8 concentrations were measured using commercially available sandwich ELISA kits according to the manufacturer’s instructions from BioLegend (San Diego, CA, USA). *n* defined as each independently treated well for a given condition, these independent samples were each run in triplicate. Data presented as mean ± SE.

### 4.5. Electrical Cell Impedance Sensing (ECIS) Wound Assay

HPAECs were grown to confluence in polycarbonate wells containing 250 μm diameter gold microelectrodes obtained from Applied BioPhysics (Troy, NY, USA). Measurements of transendothelial electrical resistance (TER) were obtained using an electric cell-substrate impedance sensing (ECIS) system obtained from Applied BioPhysics. Cells were grown on 250 μm diameter gold electrodes and baseline TER measurements were collected prior to treatment conditions, and a baseline TER of 2000 Ω was required for cells to be considered confluent on the electrodes. After treatment with addition of hemin, PBS, or ML-7, the TER was measured every 15 min for 1 h, and then cells in each microelectrode underwent ECIS-based wounding by the application of a 3 mA current at 40 kHz for 10 s. TER from each microelectrode was measured every 15 min for up to 24 h and then analyzed. TER values at each microelectrode were pooled at discrete time points and plotted versus time. *n* is defined as each independent microelectrode measurement for a given condition. Area under the curve (AUC) was calculated for each condition and presented as mean ± SE.

### 4.6. Scratch Assay

Cells were cultured on 6-well plates. Once confluent, a scratch was created across the well diameter with a sterile P-20 pipette tip followed by washout with PBS to remove cell debris. EGM with 2% FBS and treatment with either hemin, PBS, and ML-7 were added to corresponding wells. An inverted microscope with a digital camera (Nikon Eclipse TE2000-s, Nikon Instruments Inc., Melville, NY, USA) at 10× scale was used to capture the scratch at 0, 4, and 8 h. Image J (Version 1.46r, National Institutes of Health, Bethesda, Maryland, USA) was used to analyze the photos and measure the area of the created gap at each time interval, and data was ultimately represented by percentage of gap closure. *n* defined as each independently treated well for a given condition in which all time intervals were recorded. Data presented as mean gap closure percentage ± SE.

### 4.7. Immunofluorescence Cell Staining

HPAECs were grown in 35 mm glass bottom dishes and treated with control conditions, hemin, or hemin and ML-7 for 24 h. Cells were fixed in 3.7% paraformaldehyde, permeabilized with 0.25% fish skin gelatin, 0.01% saponin, 0.1% NaN_3_ in PBS, and then incubated in primary antibodies overnight. Cells were subsequently incubated with appropriate fluorochrome-conjugated secondary antibodies for 1 h. Analysis and photos were obtained using Nikon Eclipse TE2000-s inverted microscope.

### 4.8. Western Blotting

HPAECs were grown in 6-well plates and treated with 5 μM hemin, PBS, and hemin and ML-7 at different time intervals for up to 24 h. After allotted time interval, cells were washed with cold PBS and subsequently harvested by scraping the cells off the plate in RIPA buffer containing proteinase inhibitors and phosphate inhibitors per standard protocols. After centrifugation, the supernatant was collected, Laemmli buffer was added, samples were boiled for 5 min, and subsequently loaded into a 10% stacking and 3.9% separating gel. After SDS-PAGE and transfer to a nitrocellulose membrane (Bio-Rad Laboratories, Hercules, CA, USA), western blotting was performed with the appropriate primary and secondary antibodies, then visualized with chemiluminescence (Thermo Fisher Scientific, Waltham, MA, USA). *n* is defined as protein lysates extracted from each independently treated well for a given condition. Quantitative data presented as mean relative protein expression ± SE.

### 4.9. Statistical Analysis 

Statistical analysis was performed on experimental data with SigmaPlot (SigmaPlot version 11.0, SPSS Inc., Chicago, IL, USA). Student’s *t* test was performed for pairwise comparisons and analysis of variance (ANOVA) for groupwise comparisons as applicable. A repeated measures ANOVA was performed on the collected ECIS data. Values of *p* < 0.05 were considered statistically significant.

## 5. Conclusions

The direct effects of cell free hemoglobin on the pulmonary endothelium may be a crucial factor in the development and progression of precapillary pulmonary hypertension associated with chronic hemolysis. Hemin-induced endothelial dysfunction may contribute to early vascular remodeling and also work in conjunction with other mechanisms, such as nitric oxide scavenging and free radical production, to culminate in the development of pulmonary hypertension. EndoMT is identified as a novel mechanism in the development of hemin-induced endothelial dysfunction and may represent a unique therapeutic target in future studies.

## Figures and Tables

**Figure 1 ijms-23-04763-f001:**
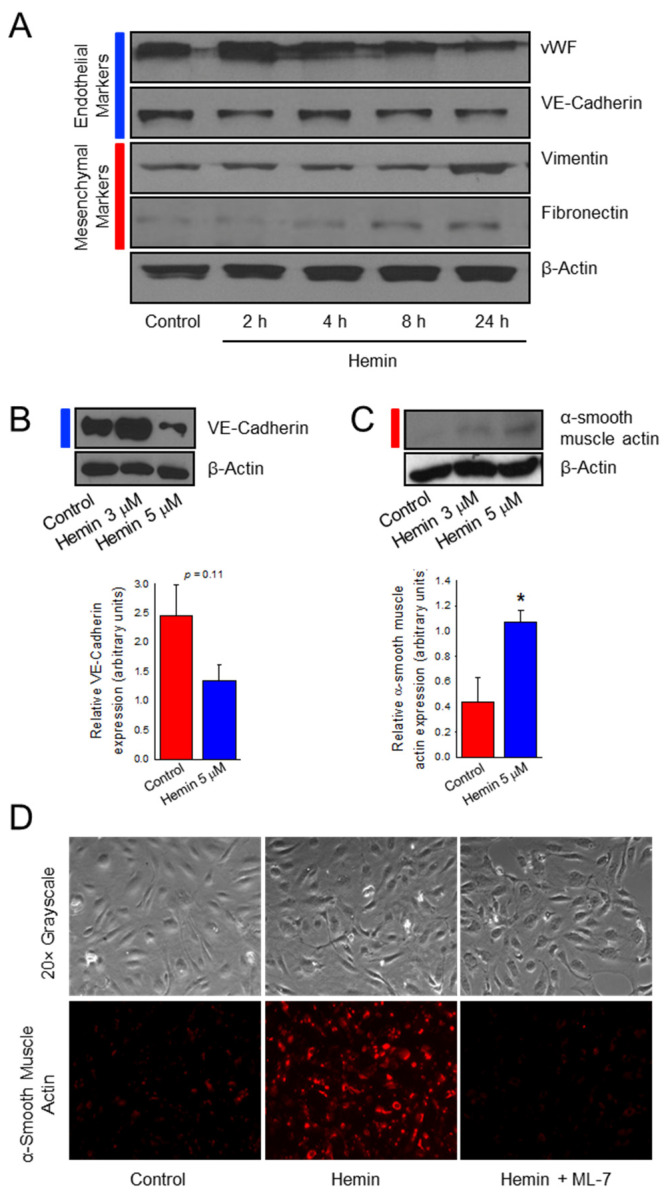
Hemin downregulates endothelial marker expression and upregulates mesenchymal marker expression in human pulmonary artery endothelial cells. (**A**) Western blot images denoting endothelial marker (blue bars) and mesenchymal marker (red bars) protein expression over time between hemin (5 μM)-treated and control (PBS vehicle)-treated HPAECs. Representative image model changes in a single experiment with reproducibility of target proteins confirmed at 8 and 24 h. Western blot images showing VE-cadherin (**B**) and α-smooth muscle actin (**C**) expression after treatment with 3 μM and 5 μM of hemin for 24 h as compared to PBS control with accompanying bar graph (*n* = 5 and 4, respectively). (**D**) Immunofluorescence images depicting α-smooth muscle actin expression after 24 h with hemin (5 μM) treatment compared to control, and pre-treatment with the myosin light chain kinase (MLCK) specific inhibitor, ML-7 (20 μM). * indicates *p* < 0.05; vWF, Von Willebrand Factor; VE-cadherin, vascular endothelial cadherin; ML-7, myosin light chain kinase specific inhibitor.

**Figure 2 ijms-23-04763-f002:**
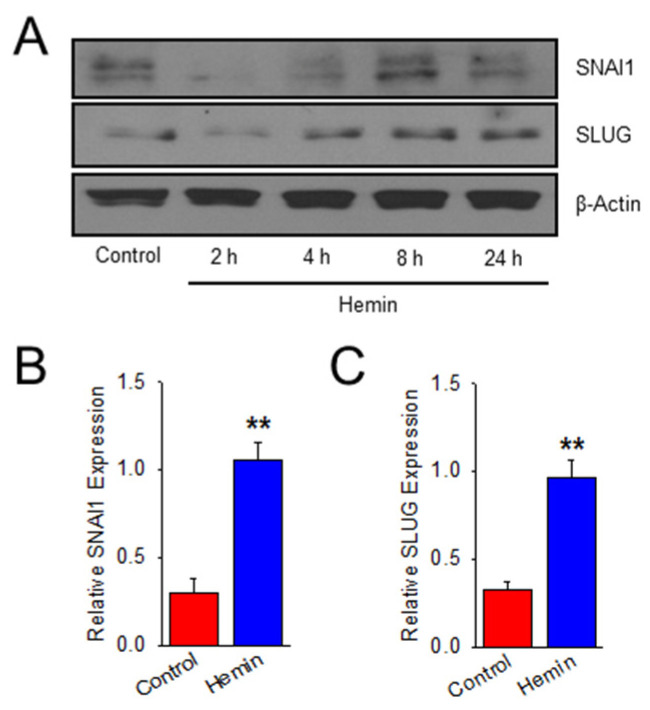
Hemin upregulates EndoMT transcription factors in human pulmonary artery endothelial cells. (**A**) Representative western blot images denoting SNAI1 and SLUG expression over time in hemin (5 μM)-treated HPAECs compared to PBS vehicle (control). Time course is constant from Figure 1 with β-actin duplicated. Bar graph summarizing relative SNAI1 (**B**) and SLUG (**C**) protein expression for hemin (5 μM)-treated HPAECs compared to PBS control at 8 h (*n* = 3). ** indicates *p* < 0.01; SNAI1, Snail homolog 1; SLUG (SNAI2), Snail homolog 2.

**Figure 3 ijms-23-04763-f003:**
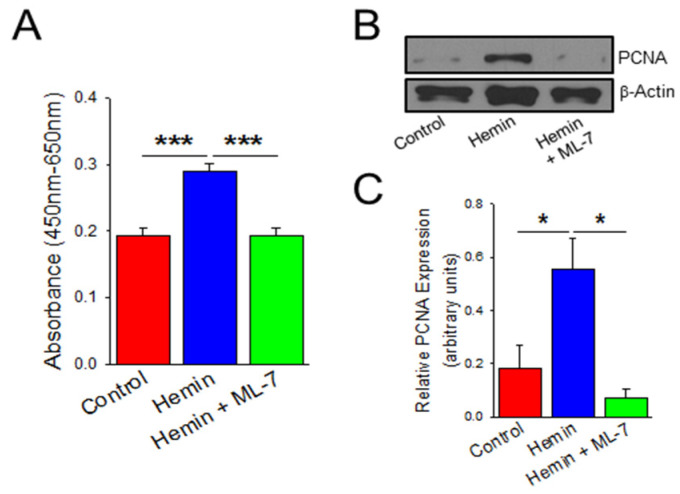
Hemin increases HPAEC proliferation which is prevented by inhibition of myosin light chain kinase (MLCK). (**A**) Bar graph depicting changes in viability by WST-1 assay in HPAECs treated for 24-h duration with hemin (5 μM), PBS vehicle, and hemin with ML-7 (20 μM) pre-treatment (*n* = 9). (**B**) Representative western blot images denoting PCNA protein expression at 24 h measured between HPAECs treated with hemin (5 μM), PBS vehicle, and hemin with ML-7 (20 μM) pre-treatment. (**C**) Bar graph summarizing relative PCNA protein expression across experiments (*n* = 6). * indicates *p* < 0.05; ***, *p* < 0.001; PCNA, proliferating cell nuclear antigen; ML-7, myosin light chain kinase specific inhibitor.

**Figure 4 ijms-23-04763-f004:**
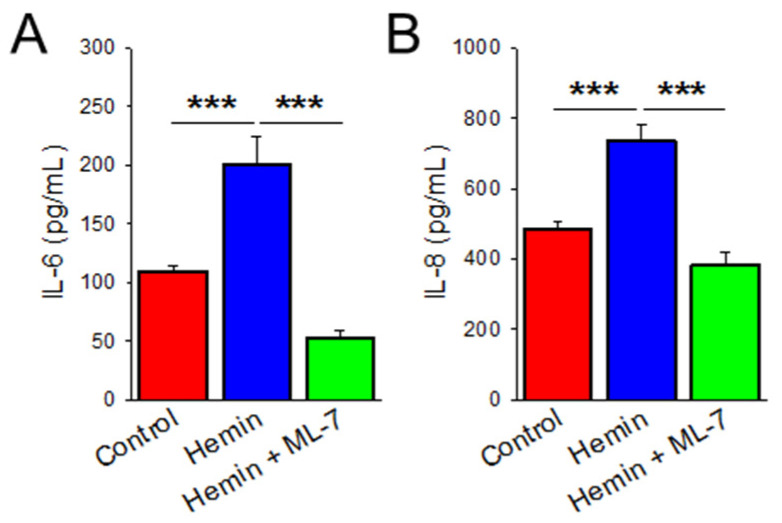
Hemin increases HPAEC cytokine release which is abrogated by MLCK inhibition. Bar graph measuring IL-6 (**A**) and IL-8 (**B**) concentrations in the media of HPAECs treated for 24-h duration with hemin (5 μM), PBS control, and hemin (5 μM) with ML-7 (20 μM) pre-treatment (*n* = 6). *** indicates *p* < 0.001; IL-6, Interleukin-6; IL-8, Interleukin-8; ML-7, myosin light chain kinase specific inhibitor.

**Figure 5 ijms-23-04763-f005:**
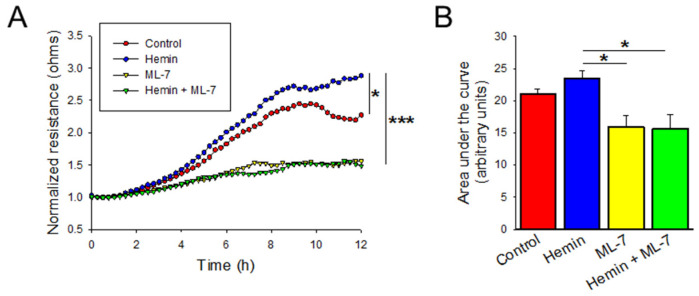
Hemin increases HPAEC migration which is abrogated by MLCK inhibition as measured by electric cell impedance sensing (ECIS). (**A**) Plot denoting the transendothelial resistance by ECIS-based wounding over time in HPAECs treated with PBS control, hemin (5 μM), PBS pre-treated with ML-7 (20 μM), and hemin pre-treated with ML-7 (20 μM) (*n* = at least 4). (**B**) Bar graph denoting area under the curve measurement for ECIS-based wounding experiments at 12 h (*n* = at least 4). * indicates *p* < 0.05; ***, *p* < 0.001; ML-7, myosin light chain kinase specific inhibitor.

**Figure 6 ijms-23-04763-f006:**
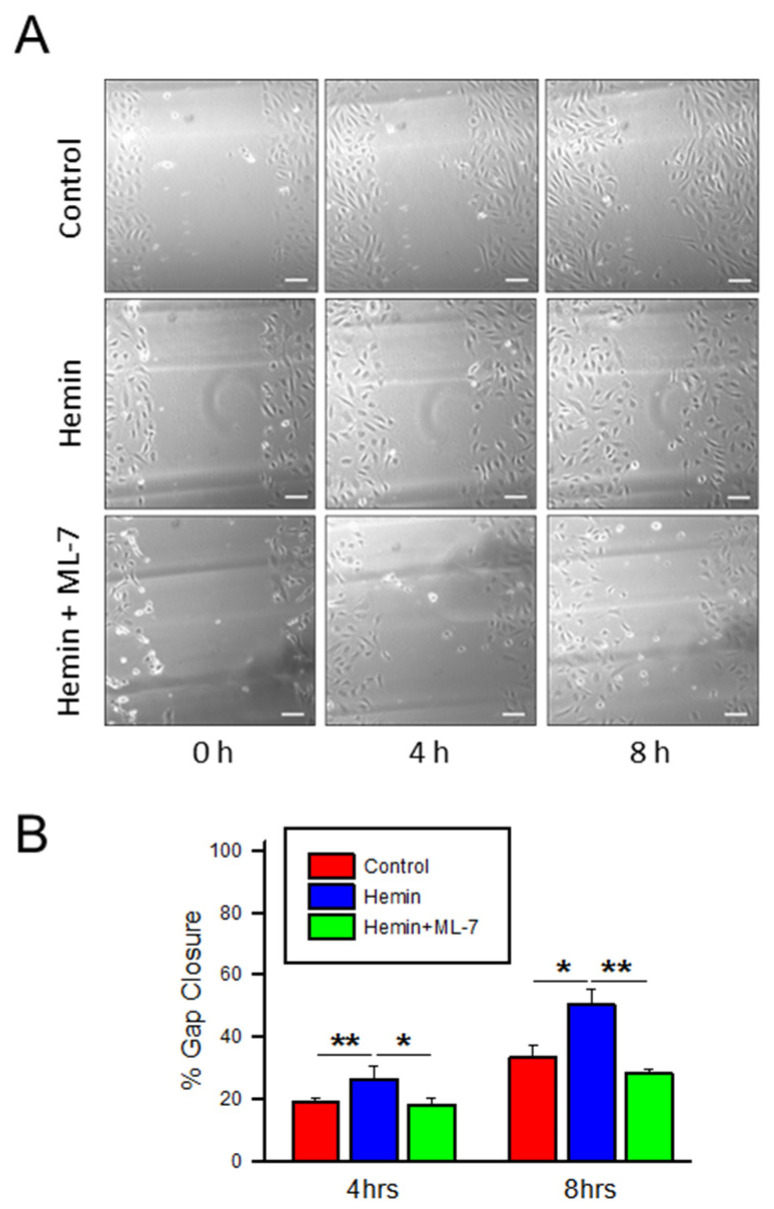
Hemin increases HPAEC migration, which is prevented by MLCK inhibition as measured by scratch wound healing assay. (**A**) Representative images of a wound healing (scratch) assay. A scratch was created in a confluent culture of HPAECs and images of wound were obtained at 0, 4, and 8 h after treatment with hemin (5 μM), PBS control, and hemin pre-treated with ML-7 (20 μM). Scale bar, 100 μm. (**B**) Bar graph denoting scratch assay gap closure at given time points (*n* = 6 for control and hemin, 4 for hemin + ML-7) * indicates *p* < 0.05; **, *p* < 0.01; ML-7, myosin light chain kinase specific inhibitor.

## Data Availability

Not applicable.

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
