# Peer review of "Hemin-Induced Endothelial Dysfunction and Endothelial to Mesenchymal Transition in the Pathogenesis of Pulmonary Hypertension Due to Chronic Hemolysis"

_ijms, 2022, doi:10.3390/ijms23094763_

Round 1
Reviewer 1 Report
The manuscript by Gonzales et al. is aimed to investigate whether hemin could induce endothelial function and EndoMT in the pathogenesis of PH due to chronic hemolysis.
Revisions made by authors are noted and acknowledged. I congratulate the authors for making the necessary revisions or providing explanations and clarifications to my previous concerns. Nonetheless, as the WST-1 assay used in this study is based on the cleavage of the tetrazolium salt WST-1 to formazan by cellular mitochondrial dehydrogenases. The larger the number of viable cells, the higher the activity of the mitochondrial dehydrogenases, and in turn the greater the amount of formazan dye formed. Therefore, I would highly suggest replacing cell proliferation with cell viability to be more accurate.
Author Response
We would like to thank the reviewer for taking the time to review our revised manuscript, “Hemin induced endothelial dysfunction and endothelial to mesenchymal transition in the pathogenesis of pulmonary hypertension due to chronic hemolysis.” We appreciate the constructive comments on the manuscript.
We agree with the reviewer that WST-1 assay is a measure of viability and proliferation, as such we have revised the results section on page 4, figure 3 legend, and description of the assay in the methods section to include viability in discussion of WST-1 assay results. Thank you for this clarification.
Reviewer 2 Report
The authors have satisfactorily responded to the feedback. I have no further comments.
Author Response
We would like to thank the reviewer for taking the time to review our revised manuscript, “Hemin induced endothelial dysfunction and endothelial to mesenchymal transition in the pathogenesis of pulmonary hypertension due to chronic hemolysis.”